

# A comparison of 1D and 2D bedload transport functions under high excess shear stress conditions in laterally-constrained gravel-bed rivers: a laboratory study

David L. Adams[1,2] and Brett C. Eaton[1]

[1]Department of Geography, University of British Columbia, Vancouver, BC, Canada
[2]School of Geography, Earth and Atmospheric Sciences, University of Melbourne, Australia

**Correspondence:** David L. Adams (dladams@alumni.ubc.ca)

**Abstract.** Channel processes under high magnitude flow events are of central interest to river science and management as they may produce large volumes of sediment transport and geomorphic work. However, bedload transport processes under these conditions remain poorly understood due to data collection limitations and the prevalence of physical models that restrict feedbacks surrounding morphologic adjustment. The extension of mechanistic bedload transport equations to gravel-bed rivers has emphasised the importance of variance in both entraining (shear stress) and resisting (grain size) forces, especially at low excess shear stresses. Using a fixed-bank laboratory model, we tested the hypothesis that bedload transport in gravel-bed rivers collapses to a more simple 1D function (i.e., with mean shear stress and median grain size) under high excess shear stress conditions. Bedload transport was well predicted by the 1D equation based on the depth-slope product, whereas a 2D equation accounting for the variance in shear stresses did not substantially improve the correlation. Back-calculated critical dimensionless shear stress values were higher for the 2D approach, suggesting that it accounts for the relatively greater influence of high shear stresses, whereas the 1D approach assumes that the mean shear stress is sufficient to mobilise the median grain size. While the 2D approach may have a stronger conceptual basis, the 1D depth-slope product approach performs unreasonably well under high excess shear stress conditions. Further work is required to substantiate these findings in laterally adjustable channels.

## 1 Introduction

The adjustment of rivers to the imposed valley gradient, sediment supply, and discharge is of central interest to geomorphology and has implications for understanding and managing natural hazards and ecological habitats. In alluvial channels, the adjustment is facilitated by the movement of bedload material, arising via the interaction between the flow and deformable boundary (Bridge and Jarvis, 1982; Dietrich and Smith, 1983; Church, 2010; Church and Ferguson, 2015). Despite there being no strict correlation between the magnitudes of perturbation and geomorphic effect (Lisenby et al., 2018), larger-than-average flows (i.e., floods) are typically associated with channel adjustment and relatievely large volumes of geomorphic work (Wolman and Miller, 1960), and extreme events may exert disproportionate control over the channel planform (Eaton and Lapointe, 2001).





The study of bedload transport processes under these relatively high discharge events is central to an understanding of river behaviour.

Researchers have dedicated considerable effort to deriving mechanistic bedload transport functions – typically empirically-calibrated – that relate the rate of movement to a force-balance between the flow and individual particles. One of the most simple and widely used relations is the Meyer-Peter and Müller (1948) equation that estimates bedload transport as a function of mean excess bed shear stress ($\bar{\tau} - \tau_c$, where $\tau_c$ is critical shear stress) for a given grain diameter, typically the median $D_{50}$. The extension of 1D bedload transport functions to gravel-bed rivers, typically characterised by a wide range of grain sizes,

necessitated several modifications that accounted for the differential mobility of grain sizes, hiding and exposure (Parker and Klingeman, 1982; Parker, 1990; Recking, 2013a; Wilcock and Crowe, 2003). Further research emphasised that at conditions where $\bar{\tau} \approx \tau_{c50}$, bedload transport is affected by the variance in shear stress (Paola and Seal, 1995; Paola, 1996; Nicholas, 2000; Ferguson, 2003; Bertoldi et al., 2009; Francalanci et al., 2012; Recking et al., 2016). More recently, Monsalve et al. (2020) proposed a 2D bedload transport function that integrates across the distribution of shear stresses, and can predict transport at lower flow conditions where $\bar{\tau} < \tau_{c50}$. In concert, these advances demonstrate a consistent trend: with decreasing

excess shear stress more information regarding grain size and shear stress (i.e., resisting and driving forces) is required to predict bedload transport.

    Considerably less is known about rivers under high relative shear stress conditions $\bar{\tau} \gg \tau_{c50}$, where most channel change occurs. This is largely due to practical limitations. Dangers associated with floods and erosion mean that researchers may collect

data before and after an event, but not during. Laboratory experiments (flumes) typically do not incorporate key degrees-of-freedom for morphologic adjustment that are available to alluvial channels, and thus do not model the full range of feedbacks between bedload transport and the deformable boundary (Church and Ferguson, 2015). Subsequently, the notion that bedload transport in rivers collapses to a more simple function (i.e., with mean shear stress and median grain size) under high excess shear stress conditions is yet to be conclusively demonstrated.Smaller-scale laboratory experiments provide an opportunity to

test this hypothesis as they model larger-scale bed and ideally bank adjustments.

    We investigate the effectiveness of 1D and 2D bedload transport functions under high relative shear stress conditions in a Froude-scaled physical model. The experiments have a widely-graded sediment mixture and develop alternate bars under pseudo-recirculating conditions at a range of widths and unit discharges. We record total bedload volumes, bathymetry, and perform 2D hydraulic modelling to apply several transport functions akin to Meyer-Peter and Müller (1948) (i.e., based on

median grain size) that capture different levels of information regarding shear stress. The results highlight the effectiveness of simple bedload transport functions under high relative shear stress in laterally constrained channels, as well as differences between 1D and 2D conceptualisations of excess shear stress and bedload transport.

## 2   Methodology

Experiments were performed in the Adjustable-Boundary Experimental System (A-BES) at the University of British Columbia

(Figure 1), a portion of which have been reported by Adams and Zampiron (2020). The A-BES comprises a 1.5 m wide





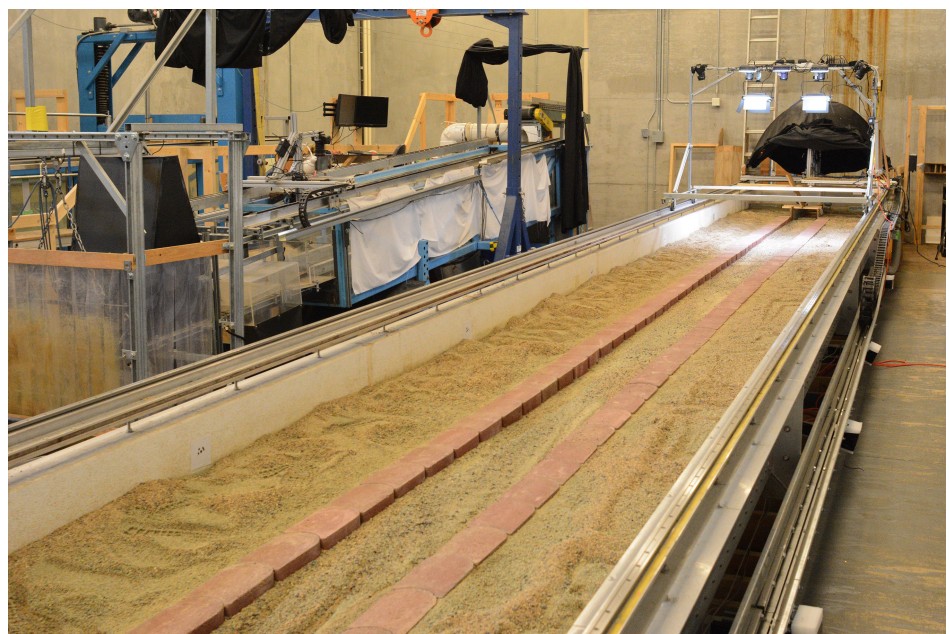

**Figure 1.** Adjustable-Boundary Experimental System (A-BES) at the University of British Columbia, featuring camera rig (top-right) and bank control system at a width of 30 cm.

**Table 1.** Summary of unit discharges $q$ ($Q/W$) used in each phase (P) of experimental Runs a-c.

| | unit discharge $q$ [L/m/s] | | | |
|---|---|---|---|---|
| | P1 | P2 | P3 | P4 |
| Run a | 5.00 | | | |
| Run b | 3.33 | | | |
| Run c | 2.22 | 3.33 | 5.00 | 7.50 |

by 12.2 m long tilting stream table, where the experiments were run as generic Froude-scaled models based on 2003 field measurements from Fishtrap Creek in British Columbia, Canada. The channel had a gradient $S$ of 0.02 m/m, average bankfull width of 10 m, formative discharge of approximately 7,500 L/s, and bulk $D_{50}$ of 55 mm. With a length scale ratio of 1:25, the A-BES was scaled to within around 30 percent of the prototype, with an initial width of 0.30 m, formative discharge $Q$ of approximately 1.5 L/s, and $D_{50}$ of 1.6 mm ($D_{84}$ = 3.2 mm, $D_{90}$ = 3.9 mm). The sediment mixture comprised natural clasts with a density of around 2,500 kg/m$^3$.

The experiments utilised interlocking landscaping bricks to constrict the channel to various widths $W$ between approximately 0.30–0.60 m. In addition to the various channel widths, four different unit discharges ($q = Q/W$) were used across the experiments (i.e., discharge was scaled by width) that increased by a factor of 1.5 (Table 1). Two constant-discharge runs





**Table 2.** Summary of experiments conducted in the A-BES. DEM count excludes screeded bed. Experiment 1 is published in Adams and Zampiron (2020).

| Exp. | W [m] | Q [L/s] | Duration [hrs] | DEMs |
|------|-------|---------|----------------|------|
| Exp1a | 0.30 | 1.50 | 16 | 24 |
| Exp1b | 0.30 | 1.00 | 16 | 24 |
| Exp1c | 0.30 | 0.66, 1.00, 1.50, 2.25 | 8, 4, 4, 4 | 20, 16, 16, 16 |
| Exp3a | 0.45 | 2.25 | 16 | 24 |
| Exp3b | 0.45 | 1.50 | 16 | 24 |
| Exp3c | 0.45 | 1.00, 1.50, 2.25, 3.37 | 8, 4, 4, 4 | 20, 16, 16, 16 |
| Exp4a | 0.60 | 2.00 | 16 | 24 |
| Exp4b | 0.60 | 3.00 | 16 | 24 |
| Exp4c | 0.60 | 1.33, 2.00, 3.00, 4.50 | 8, 4, 4, 4 | 20, 16, 16, 16 |

used the middle two discharges, and one multi-discharge run consisted of the four discharges in increasing order. A full list of experiments is provided in Table 2.

At the beginning of each experiment the bulk mixture was mixed by hand to minimise lateral and downstream sorting, and then the in-channel area was screeded to the height of weirs at the upstream and downstream end using a tool that rolled along the brick surface. The flow was run at a low rate with little-to-no movement of sediment until the bed was fully saturated, and was then rapidly increased to the target flow.

Three different types of data were collected throughout each experiment; surface photos, stream gauge measurements, and sediment output. A rolling camera rig positioned atop the A-BES consisted of five Canon EOS Rebel T6i DSLRs with EF-S 18–55 mm lenses positioned at varying oblique angles in the cross-stream direction to maximise coverage of the bed, and five LED lights. Photos were taken in RAW format at 0.2 m downstream intervals, providing a stereographic overlap of over two-thirds. Ten water stage gauges comprised of a measuring tape (with 2 mm intervals) on flat boards were located along the inner edge of the bricks every 1 m. To minimise edge effects, gauges were not placed within 0.60 m of either the inlet or the outlet. The gauges were read at an almost horizontal angle which, in conjunction with the dyed blue water, minimised systematic bias towards higher readings due to surface tension effects. Based on the measurement precision of the stream gauge readings, errors of 6–11 percent could be expected for mean hydraulic depths ($h = A_c/w$, where $A_c$ is flow cross-sectional area and $w$ is wetted width).

The data collection procedure was designed to maximise measurement accuracy as much as reasonably possible. Given that stream gauge data would later be paired with topographic data, the timing of gauge readings needed to closely coincide with surface photography. Every time photos were taken the bed was drained, as the surface water would distort the photos. These constraints necessitated a procedure in which manual stream gauge readings (to the nearest 1 mm) were taken 30–40 seconds before the bed was rapidly drained, around the minimum time it would take to obtain the readings. The bed was then photographed and gradually re-saturated before resuming the experiment, approximately 10 minutes.





Each experimental phase was divided into a series of segments between which the data was collected. The procedure occurred in 5, 10, 15, 30, 60, and 120 minute segments with four repeats of each (i.e., 4 x 5 min, 4 x 10 min), which was designed to reflect the relatively rapid rate of morphologic change at the beginning of each phase. For example, in wider channels, alternate
bars developed within an hour, and there was relatively little morphologic change in the following hours (Adams and Zampiron, 2020; Adams, 2021).

Throughout the experiments, sediment falling over the downstream weir was collected in a mesh bucket, drained of excess water, weighed damp to the nearest 0.2 kg, placed on the conveyor belt at the upstream end, and gradually recirculated at the same rate it was output, as opposed to a 'slug' injection. Based on a range of samples collected across the experiments, we
determined the weight proportion of water to be approximately 5.8 percent and applied this correction factor to obtain approximate dry weights. There was no initial feed of sediment, although this no-feed period was only 5 minutes. The experiments are best described as pseudo-recirculating as sediment was fed at the end of each segment, and every 15 minutes, regardless of whether the bed was drained.

## 2.1 Data processing

Using the images, point clouds were produced using structure-from-motion photogrammetry in Agisoft MetaShape Professional 1.6.2 at the highest resolution, yielding an average point spacing of around 0.25 mm. Twelve spatially-referenced control points and additional unreferenced ones were distributed throughout the A-BES, which placed photogrammetric reconstructions within a local coordinates system and aided in the photo-alignment process. Using inverse distance weighting, the point clouds were converted to digital elevation models (DEMs) at 1 mm horizontal resolution. Despite the use of control points, the
DEMs contained a slight arch effect in a downstream direction whereby the middle of the model was bowed upwards, which was an artefact of the photogrammetric reconstruction. This effect was first quantified by applying a quadratic function along the length of the bricks, which represent an approximately linear reference elevation (brick elevations vary by $\pm$ 4 mm). The arch was then removed by determining correction values along the length of the DEM using the residuals, which were then applied across the width of the model.

For each DEM, ten wetted cross-sections were reconstructed using the water surface elevation data, which were then used to estimate reach-averaged hydraulics. For more detailed spatial analysis, the flow conditions of water depth, and shear stress were reconstructed using a 2D numerical flow model (Nays2DH) to the final DEM of each discharge phase. Nays2DH is a two-dimensional, depth-averaged, unsteady flow model that solves the Saint-Venant equations of free surface flow with finite differencing based on a general curvilinear coordinate system (further details can be found in Nelson et al., 2016). Key input
boundary conditions are the channel DEM, an initial estimate of reach-averaged Manning's n, cell resolution, and the water discharge. We selected an n value of 0.045 based on the channel conditions, a cell resolution equivalent to 5 mm, and a flow duration of 100 s was sufficient to establish convergence. After initially estimating n, we back-calculated a spatially variable value using the flow resistance law presented by Ferguson (2007), and ran the solver again.

To minimise rounding errors associated with the relatively shallow depths in our experiments, the DEM size and discharge
were adjusted to the prototype scale (i.e., using a length scale ratio of 25). The estimated water depths, shear stresses and



velocities from Nays2DH were then back-transformed to the model scale (Table 3). We removed cells with relatively shallow flows defined as depths less than $2D_{84}$ as they contributed a large peak in the frequency distribution of flow depths and likely account for a small proportion of bedload activity. We define areas of the bed with flows above this threshold as 'wetted'. The mean-normalised frequency distributions of flow depths and shear stresses were fitted with gamma and Gaussian distributions

(coefficients in Table 3), where the goodness-of-fit was assessed using both Kolmogorov-Smirnov and Anderson-Darling tests.

The results of the flow model were quantitatively validated by comparing measured reach-averaged hydraulic depths to modelled ones (Figure 2). Most estimates fall within 10–15 percent of the line of equality, although the flow model estimates a narrower range (approximately 12–18 mm) of mean hydraulic depths across the experiments compared to the stream gage measurements (11–21 mm). Stream gages are easily biased towards deep or shallow flows due to there being only ten fixed

points, thus explaining the wider range of the estimates. The stream gages only serve as an approximation to validate the flow model.

**Table 3.** Summary of mean experimental and flow model results. Parameters $w$ = wetted width [m], $d$ = flow depth [m], $U$ = velocity [m/s], Fr = Froude number, Re = Reynolds number ($Ud/v$, where $v$ is the kinematic viscosity), $\bar{\tau}$ = mean shear stress [Pa], $q_b$ = unit bedload transport [kg/m/min], $\sigma_\tau$ is the standard deviation of shear stress, $\alpha$ and $\beta$ parameters describe the fitted gamma distribution of shear stress. The parameters A1, A2, B1, B2 refer to the four approaches outlined in Table 4.

| Exp | $w$ | $d$ | $w/d$ | $U$ | Fr | Re | $\bar{\tau}$ | $q_b$ | A1 | A2 | A3 | A4 | $\sigma_\tau$ | $\alpha$ | $\beta$ |
|---|---|---|---|---|---|---|---|---|---|---|---|---|---|---|---|
| Exp1a | 0.26 | 0.015 | 17.2 | 0.36 | 0.92 | 4117 | 2.65 | 1.75 | 1.02 | 0.54 | 0.94 | 0.70 | 0.46 | 3.80 | 0.26 |
| Exp1b | 0.21 | 0.013 | 15.9 | 0.31 | 0.83 | 3132 | 2.34 | 0.97 | 0.58 | 0.38 | 0.45 | 0.50 | 0.50 | 3.15 | 0.32 |
| Exp1c(1) | 0.18 | 0.012 | 15.0 | 0.28 | 0.80 | 2528 | 2.04 | 0.34 | 0.24 | 0.18 | 0.17 | 0.24 | 0.49 | 3.04 | 0.33 |
| Exp1c(2) | 0.21 | 0.013 | 16.6 | 0.30 | 0.81 | 2995 | 2.25 | 0.86 | 0.47 | 0.34 | 0.38 | 0.68 | 0.53 | 2.41 | 0.42 |
| Exp1c(3) | 0.26 | 0.016 | 16.5 | 0.34 | 0.87 | 4116 | 2.81 | 1.81 | 1.30 | 0.71 | 1.11 | 0.87 | 0.46 | 3.60 | 0.28 |
| Exp1c(4) | 0.28 | 0.018 | 15.1 | 0.44 | 1.03 | 6118 | 3.36 | 3.68 | 2.39 | 1.21 | 2.14 | 1.08 | 0.39 | 5.79 | 0.17 |
| Exp3a | 0.37 | 0.015 | 25.3 | 0.34 | 0.87 | 3869 | 2.69 | 2.48 | 1.10 | 0.69 | 0.84 | 0.95 | 0.49 | 2.79 | 0.36 |
| Exp3b | 0.28 | 0.014 | 20.0 | 0.33 | 0.87 | 3609 | 2.45 | 1.10 | 0.72 | 0.39 | 0.67 | 0.74 | 0.47 | 3.09 | 0.32 |
| Exp3c(1) | 0.23 | 0.013 | 17.9 | 0.30 | 0.83 | 3010 | 2.17 | 0.53 | 0.38 | 0.20 | 0.42 | 0.32 | 0.46 | 3.24 | 0.31 |
| Exp3c(2) | 0.29 | 0.013 | 22.1 | 0.31 | 0.83 | 3173 | 2.35 | 1.21 | 0.59 | 0.36 | 0.45 | 0.70 | 0.49 | 3.04 | 0.33 |
| Exp3c(3) | 0.36 | 0.015 | 23.6 | 0.35 | 0.89 | 4083 | 2.69 | 2.01 | 1.10 | 0.70 | 0.96 | 1.11 | 0.50 | 2.54 | 0.39 |
| Exp3c(4) | 0.40 | 0.017 | 23.5 | 0.41 | 0.97 | 5405 | 3.21 | 4.34 | 2.09 | 1.30 | 1.68 | 1.61 | 0.47 | 3.30 | 0.30 |
| Exp4a | 0.48 | 0.015 | 32.2 | 0.35 | 0.89 | 3966 | 2.74 | 2.42 | 1.18 | 0.74 | 0.85 | 1.03 | 0.50 | 2.90 | 0.34 |
| Exp4b | 0.40 | 0.013 | 29.6 | 0.31 | 0.81 | 3194 | 2.28 | 1.14 | 0.50 | 0.36 | 0.50 | 0.83 | 0.53 | 2.15 | 0.46 |
| Exp4c(1) | 0.31 | 0.013 | 23.5 | 0.31 | 0.84 | 3114 | 2.11 | 0.49 | 0.31 | 0.17 | 0.45 | 0.41 | 0.49 | 2.45 | 0.41 |
| Exp4c(2) | 0.39 | 0.014 | 28.2 | 0.32 | 0.85 | 3398 | 2.33 | 1.22 | 0.56 | 0.35 | 0.58 | 0.78 | 0.50 | 2.94 | 0.34 |
| Exp4c(3) | 0.46 | 0.015 | 29.9 | 0.37 | 0.93 | 4390 | 2.80 | 2.52 | 1.28 | 0.67 | 1.05 | 1.02 | 0.45 | 2.82 | 0.35 |
| Exp4c(4) | 0.57 | 0.018 | 32.1 | 0.42 | 0.97 | 5660 | 3.15 | 4.41 | 1.96 | 0.98 | 1.89 | 1.67 | 0.40 | 5.15 | 0.19 |




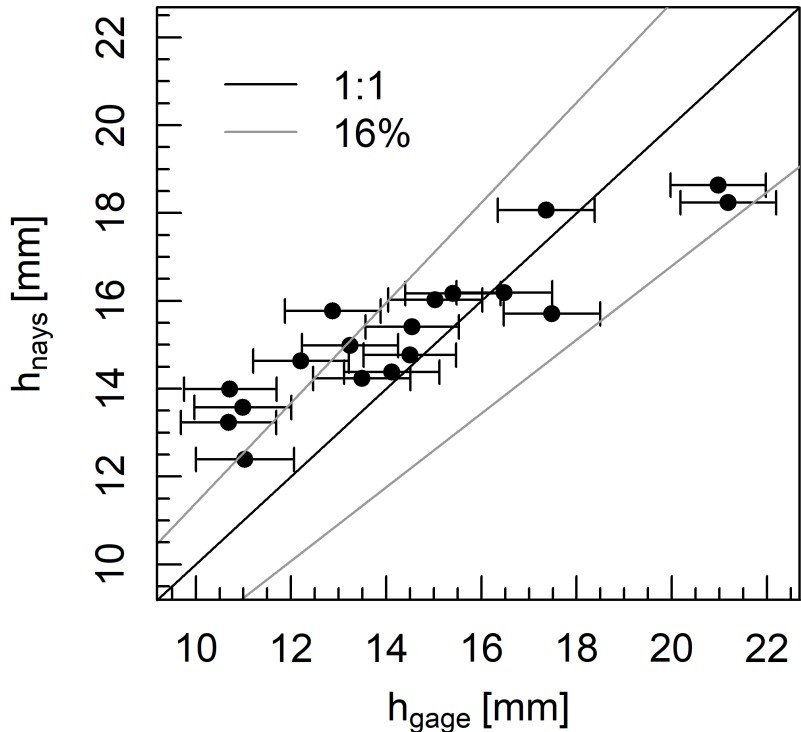

**Figure 2.** Measured versus modelled mean hydraulic depth $h$ at the end of each experimental phase, featuring 16 percent bounds.

## 2.2 Determining a representative sediment transport rate

The channels are formed under constant discharge conditions for 4–16 hours, beginning from either a screeded bed or a morphology developed at a lower discharge. Each experimental phase comprises an initial adjustment period during which morphology, hydraulics, and sediment transport are non-stationary. This adjustment period, which may vary from minutes to an hour, is followed by a steady-state period where these characteristics fluctuate around a mean value (see Adams, 2020; Adams and Zampiron, 2020). Under recirculating conditions, the stationarity of bedload transport represents a condition in which there is no net aggradation or degradation over time. In Figure 3 we present two typical examples of sediment transport fluctuations under constant conditions for 16 hours. In both examples, there is a brief adjustment period with less sediment





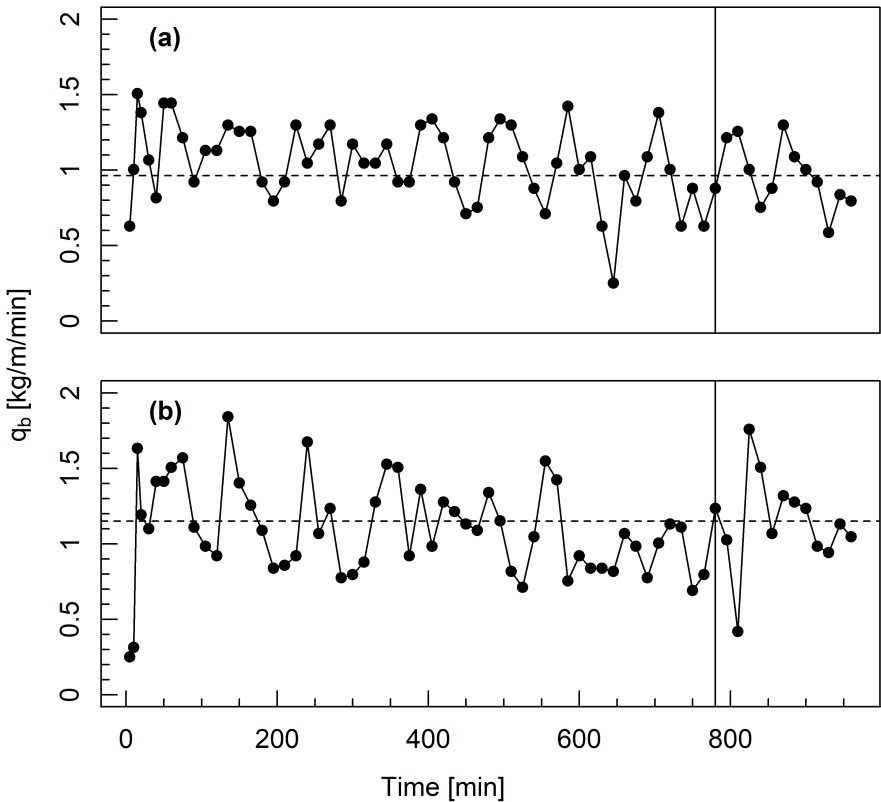

**Figure 3.** Width-averaged bedload transport over time in two experiments with different widths but similar reach-averaged shear stress: a) Experiment 1b (W = 0.30 m, $\bar{\tau}$ = 2.34 Pa), and b) Experiment 4b (W = 0.60 m, $\bar{\tau}$ = 2.28 Pa). The beginning of the time window over which bedload transport is averaged is indicated by the solid vertical line, and mean transport over this period is indicated by a horizontal dashed line.

transport, followed by fluctuations around a mean value. We consider these fluctuations as being associated with second-order processes such as bar reshaping and sediment waves (e.g., Dhont and Ancey, 2018), which are outside the scope of this study.

We determined a representative sediment transport rate for each experimental phase by averaging output over the final three-hour period (Table 3), thus removing the initial adjustment period. There is little difference between averaging over the final hour versus the final three hours, with almost all average values being ± 12.5 percent. There were three instances where these

two averaging windows yielded values differing by 15–25 percent due to high-magnitude fluctuations around an otherwise stationary bedload transport rate.





## 2.3 1D and 2D excess shear stress

We examined the correlation between the observed representative sediment transport rate and two formulations of excess shear stress based on the Meyer-Peter and Müller (1948) equation

$$q_b = k(\bar{\tau} - \tau_c)^{1.6} \tag{1}$$

where $q_b$ is width-averaged bedload transport, $k$ accounts for flow resistance and the relative density of sediment, and the exponent 1.6 is based on Wong and Parker (2006). The value of $k$ is highly variable across empirical datasets, whereas the exponent is relatively consistent (Gomez and Church, 1989). The critical shear stress value for the $D_{50}$ ($\tau_{c50}$) is estimated by $\tau_c^* g(\rho_s - \rho)D$, where $\tau_c^*$ is the dimensionless critical shear stress, $g$ is gravity, $\rho$ is the density of water, $\rho_s$ is the density of sediment

We aimed to investigate the concepts underlying 1D and 2D bedload transport equations, rather than refine them. Subsequently, we ignored the parameter $k$ that typically varies across channels and simplify Equation 1 to express the correlation between observed sediment transport and mean excess shear stress (raised to the exponent):

$$q_b \propto (\bar{\tau} - \tau_{c50})^{1.6} \tag{2}$$

This equation was modified to integrate across the distribution of local shear stresses

$$q_b \propto \int (\tau_{(x)} - \tau_{c50})^{1.6} dx/A \tag{3}$$

where $\tau_{(x)}$ is local bed shear stress and $A$ is the total bed area. Equations 2 and 3 are 1D and 2D approaches to correlating observed transport capacity with excess shear stress. We applied both equations using shear stress values calculated in two ways: 1) depth-slope product ($\tau = \rho g d S$), and 2) 2D flow modelling, thus yielding four different approaches (Table 4). We intentionally did not account for sinuosity or side-wall effects in the former. In the case of the 1D depth-slope approach, depth was calculated using the mean depth and mean channel gradient, whereas in the 2D depth-slope we varied depth but the gradient remained constant. For each approach, we back-calculated the optimal value of $\tau_c^*$ by systematically varying it and finding the strongest correlation (least-squares linear fit) between $q_b$ and excess shear stress (i.e., $[\bar{\tau} - \tau_{c50}]^{1.6}$ or $\Sigma[\tau_x - \tau_{c50}]^{1.6}/A$), indexed by root-mean-square-error (RMSE), which is shown in Figure 4. We report optimised values of $\tau_c^*$ and least-squares goodness-of-fit statistics in Table 4, and also include values obtained using the exponent 1.5 in each equation.





**Table 4.** Optimised values of $\tau_c^*$ and goodness-of-fit statistics for correlations between excess shear stress and observed bedload transport using four different approaches. Values obtaining using the exponent 1.5 are presented in parentheses, and $\bar{\tau}/\tau_{c50}$ represents the range of relative shear stress values across the experiments.

| Approach | Equation | $\tau$ method | $\tau_c^*$ | r$^2$ | RMSE | $\bar{\tau}/\tau_{c50}$ |
|----------|----------|---------------|------------|-------|------|-------------------------|
| A1 | 2 (1D) | d/S | 0.077 (0.079) | 0.96 | 0.43 (0.42) | 1.17–1.81 |
| A2 | 3 (2D) | d/S | 0.104 (0.107) | 0.94 | 0.52 (0.52) | 0.24–1.67 |
| B1 | 2 (1D) | modelled | 0.063 (0.066) | 0.97 | 0.35 (0.35) | 1.25–2.06 |
| B2 | 3 (2D) | modelled | 0.107 (0.110) | 0.97 | 0.36 (0.35) | 0.17–1.30 |

## 3  Results

Under the imposed channel widths (0.30–0.60 m) and unit discharges (2.22–7.50 L/m/s) all channels developed an alternate bar morphology with pools, bars, and riffles (see Figure 5 for an example). Especially at low unit discharges, wetted areas ($d > 2D_{84}$) on average occupied a relatively small portion of the total available width, between 52 and 95 percent. When unit discharge was calculated using the wetted width, it was closely correlated with mean shear stress based on least-squares linear regression (Figure 6a). Using the 1D approach, the depth-slope method estimated higher values of mean shear stress (5–23 percent higher) compared to the numerical model, and also higher values of critical dimensionless shear stress ($\tau_c^* = 0.077$ and 0.063, respectively, Table 4). Both methods yielded similar estimates of excess shear stress ($\bar{\tau}$ / $\tau_{c50} = 1.17–1.81$ and 1.25–2.06, respectively).

Estimated values of $\tau_c^*$ using the 2D approach were consistently higher than the values obtained using the 1D approach, but were less sensitive to how shear stress was calculated ($\tau_c^* \approx 0.105$ for both methods). Based on the 2D approach, the proportion of the bed area experiencing excess shear stress was linearly related to unit discharge and ranged between 18–70 percent (Figure 6b). In several experiments 2D estimates of $\tau_{c50}$ were higher than $\bar{\tau}$. Shear stresses at or below the mean were estimated to exceed $\tau_{c50}$ only at the highest unit discharges ($q > 5.5$ L/m/s), accounting for 10–25 percent of the total bed area, or 18–33 percent of the area where $\bar{\tau} > \tau_{c50}$. We further visualise shear stress distributions and estimated critical values using examples in Figure 7b.

Frequency distributions of normalised flow depth and shear stress followed both Gaussian and gamma distributions (Figure 7a), confirmed by both Kolmogorov-Smirnov and Anderson-Darling tests ($p < 0.1$). These distributions are qualitatively similar based on their cumulative distributions following the removal of shallow depths. In the case of the shear stress distributions, the shape parameter $\alpha$ was linearly related to unit discharge based on least-squares regression (RMSE = 0.69, r$^2$ = 0.39, $p < 0.01$), and the scale parameter $\beta$ was negatively correlated (RMSE = 0.58, r$^2$ = 0.32, $p < 0.01$). The parameters of the gamma distribution indicate that with increasing unit discharge the distribution of shear stress became more concentrated and less positively skewed.





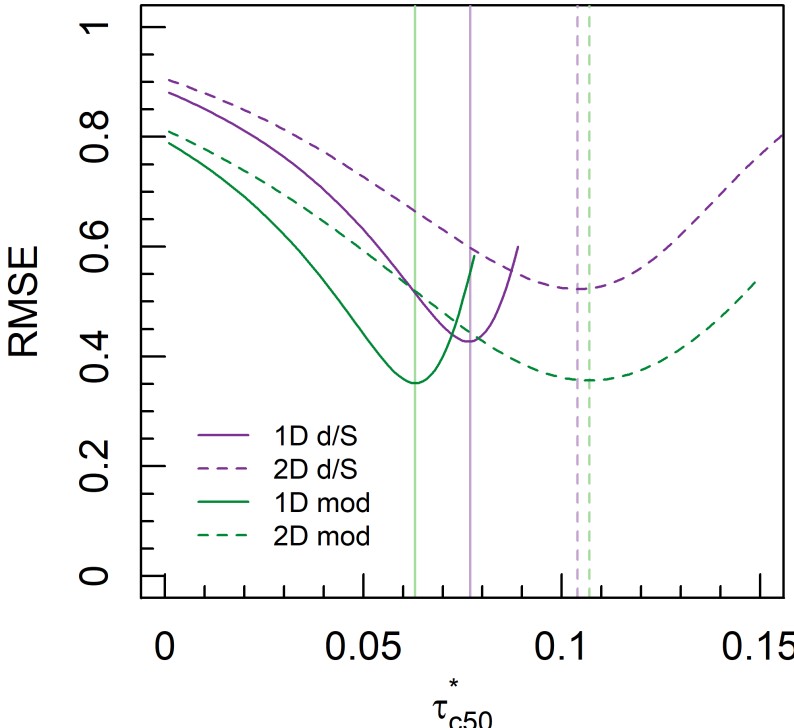

**Figure 4.** The correlation between $q_b$ and excess shear stress (indexed by RMSE) with varying critical dimensionless shear stress for each approach. Back-calculated critical dimensionless value is indicated where RMSE is lowest (Table 4).

Despite following similar frequency distributions, local flow depth and shear stress were not strongly coupled spatially
(Figure 8). These two parameters are roughly correlated but with considerable scatter, whereby for a given grid cell normalised shear stress is commonly more than a factor-of-two greater or less than normalised flow depth. The spatial decoupling of flow depth and shear stress is also evident in Figure 5, especially where areas of high shear stress are estimated to occur immediately downstream of pools where flow is deepest.

We present the correlation between bedload transport and the four different representations of excess shear stress in Figure 9.
These represent combinations of two different methods of calculating bed shear stress, depth-slope product and numerically modelled, against 1D and 2D representations of excess shear stress (Table 4). All four methods yield similar correlations



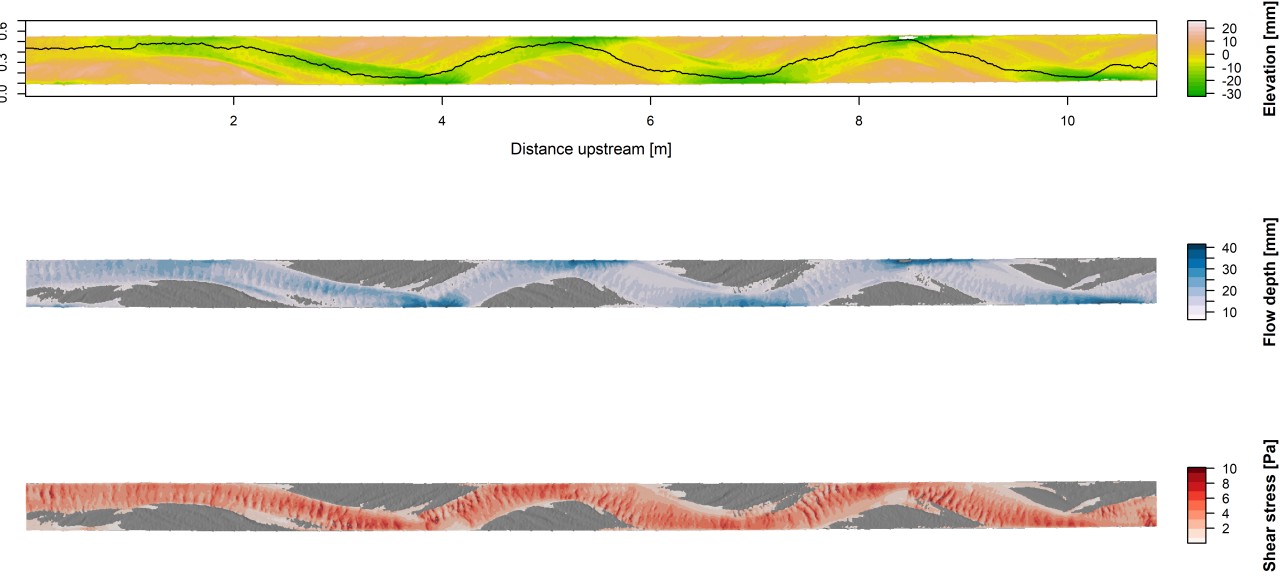

**Figure 5.** Channel area at the conclusion of Experiment 3b ($W = 0.45$ m, $\bar{\tau} = 2.41$ Pa) displaying characteristics (top to bottom): a) elevation, b) flow depth, and c) shear stress from flow model. Cells where $d < 2D_{84}$ are not shown. Transect along path of highest bed shear stress is displayed as a black line.

between excess shear stress and observed bedload transport, indicated by RMSE values between 0.35 and 0.52, where these end-values correspond to the 1D modelled shear stress (B1) and 2D depth-slope product approach (A2), respectively. Changing the exponent from 1.6 to 1.5 in Equations 2 and 3 had almost no effect on the estimated values of $\tau_c^*$ or the prediction errors.

## 4 Discussion

These experiments have several advantages over traditional field and flume datasets. Although the experiments do not model lateral adjustment, the smaller scale-ratio (1:25) means they incorporate morphology and processes at a larger scale compared to most flumes with width-ratios between approximately 15 and 40. The bulk mixture comprises a wide range of grain sizes (0.5–8.0 mm) that have been demonstrated to modulate channel adjustment, especially under conditions where the larger-than-average grain size is only partially mobile (MacKenzie and Eaton, 2017; MacKenzie et al., 2018; Booker and Eaton, 2020; Adams, 2021). We measured total bedload volumes and adjustments to bed topography during flood stages, which is not possible in the field or in many recirculating experiments. The applied flows are longer and more constant than floods typically observed in nature ($> 20$ hours in the field prototype), which allows the experiments to reach an idealised steady-state whereby morphology, hydraulics, and bedload fluctuate around a mean condition (Figure 3). These characteristics make the





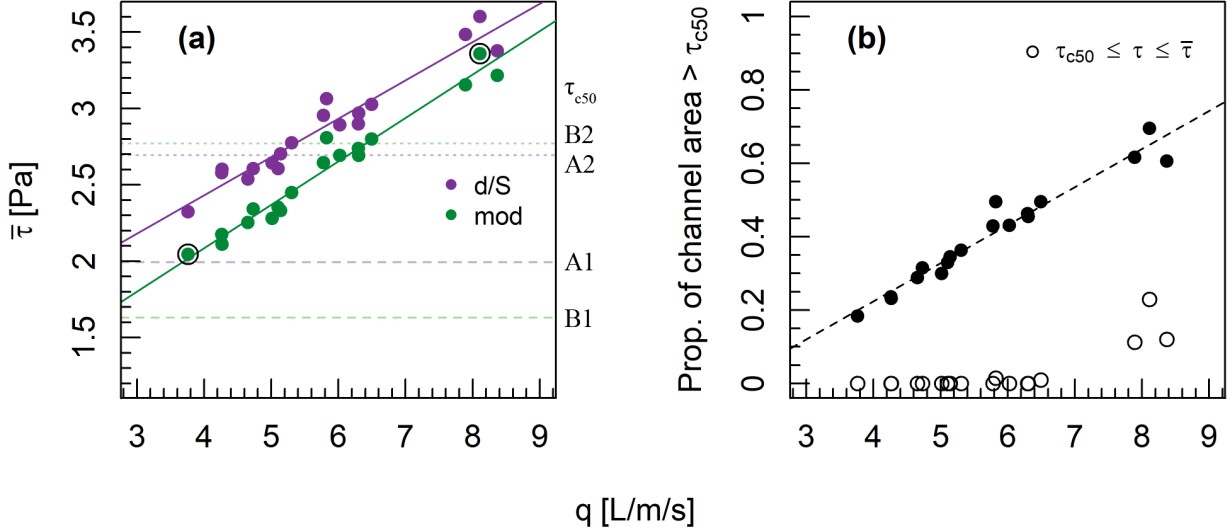

**Figure 6.** (a) Relationship between unit discharge (calculated using wetted width) and mean shear stress using depth-slope product (RMSE = 0.097, $r^2$ = 0.93, $p < 0.001$) and modelled shear stresses (RMSE = 0.073, $r^2$ = 0.96, $p < 0.001$). Horizontal lines indicate fitted values of $\tau_{c50}$, and circled points indicate channels with the highest and lowest shear stress used in Figure 7b. (b) Relationship between unit discharge and the proportion of the wetted channel area ($d > 2D_{84}$) where $\tau > \tau_{c50}$ using modelled shear stresses, as well as the proportion of channel area where $\tau_{c50} < \tau \leq \bar{\tau}$. Least-square linear regression is indicated by a dashed line (RMSE = 0.030, $r^2$ = 0.95, $p < 0.001$).

experimental dataset appropriate for investigating the effectiveness of spatially- and temporally-averaged (double-averaged) bedload transport equations in laterally-constrained gravel-bed rivers under high relative shear stress conditions.

We evaluated four different bedload transport functions based on the correlation between excess shear stress and observed volumes of bedload transport, averaged over the final three hours of each experimental phase. We first focus our discussion on three of these approaches in increasing order of sophistication (A1, B1, then B2), and then explain their relative effectiveness.
Finally, we discuss the conceptual differences between 1D and 2D bedload transport functions.

### 4.1   Comparison between prediction errors

Most bedload transport functions index the applied excess shear stress using the mean depth-slope product as this data is relatively easy to collect in field contexts (Gomez and Church, 1989; Barry et al., 2004; Recking, 2013b). This approach relies on the assumption that local variations in channel gradient and flow depth cancel out, such that mean flow depth is





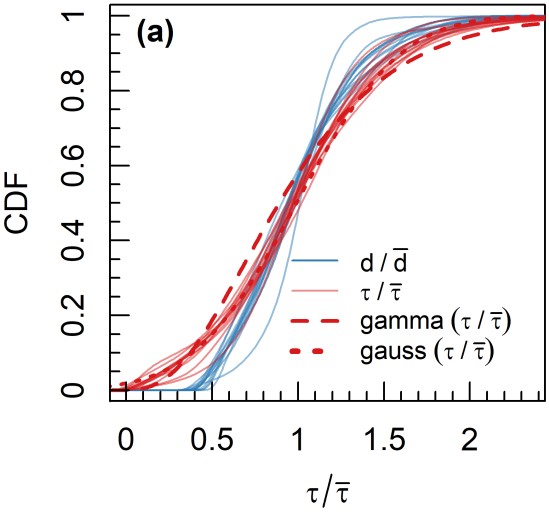
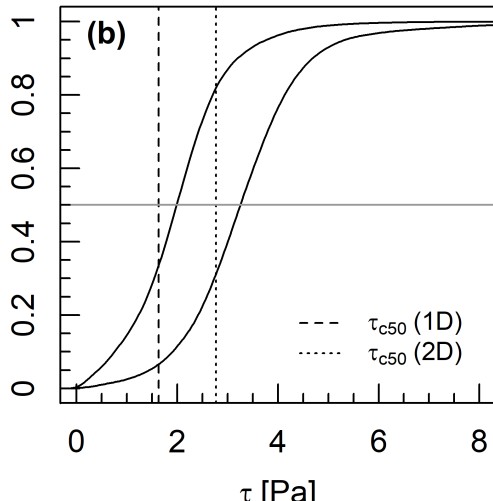

**Figure 7.** a) Cumulative frequency distributions of normalised (modelled) flow depth and shear stress at the end of each experimental phase, where the upper end of the distribution has been truncated to approximately the 99th percentile. Note the absence of shallow depths ($d < 2D_{84}$). The average gamma distribution fit for the normalised shear stress distribution is included ($\alpha = 3.30$, $\beta = 0.30$), as well as the average Gaussian fitted distribution ($\sigma = 0.47$). b) Cumulative frequency distribution of non-normalised modelled shear stresses in experimental phases with highest (Exp1c(1)) and lowest (Exp1c(4)) mean shear stress (circles in Figure 6). Estimates of $\tau_{c50}$ using 1D and 2D approaches (B1 and B2, respectively) are indicated by dashed lines, and the horizontal line is the median shear stress that closely corresponds to the mean.

proportional to mean shear stress (Nicholas, 2000; Ferguson, 2003). We do indeed observe this condition whereby normalised flow depth and shear stress follow similar frequency distributions (Figure 7a), despite being decoupled spatially (Figure 8). The approach A1 (1D depth-slope product) in our analysis is the most similar to the traditional field approach. It is even more idealised as it did not account for sinuosity (note the slight sinuosity in Figure 5 that reduces the mean channel gradient), flow resistance, or energy losses to the channel banks. The strength of the correlation between excess shear stress and bedload transport (RMSE = 0.43) provides a reference point for other approaches.

In recent decades, technological advancements in remote sensing and hydraulic modelling have allowed researchers to directly model bed shear stress, thus providing a potentially more accurate estimate. This advancement is utilised in the B1 approach (1D modelled shear stress), which accounts for the effect of both sinuosity, flow resistance, and energy losses to the channel banks. Accounting for these additional factors may explain the 19 percent reduction in RMSE (0.35) compared



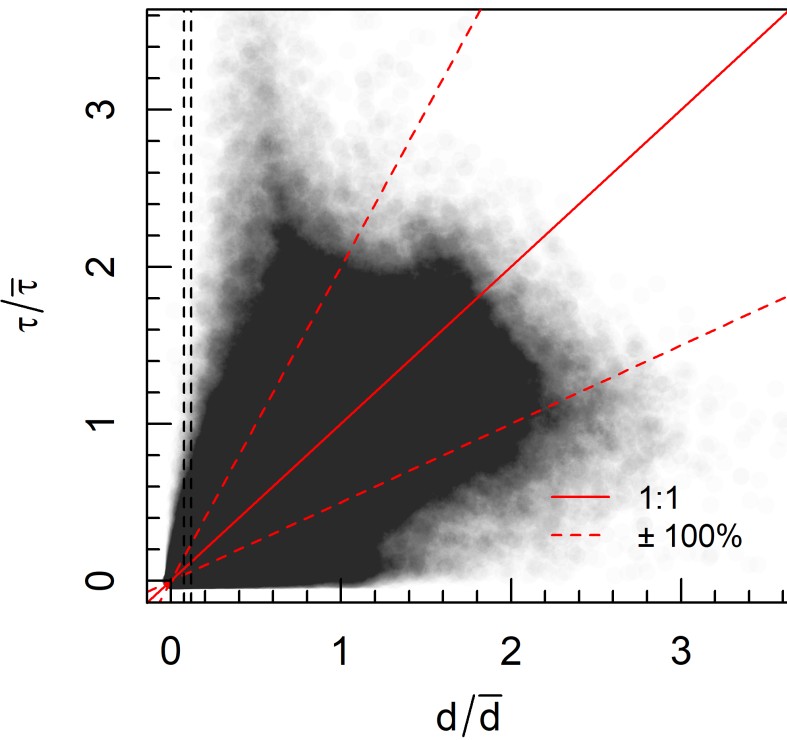

**Figure 8.** Relationship between local normalised flow depth and shear stress across all experiments, produced by randomly sampling 10 percent of cells from each flow model. Vertical dashed lines indicate the range of normalised flow depth that were used to threshold the flow model.







**Figure 9.** Correlation between excess shear stress and observed bedload transport using the four approaches outlined in Table 4. The dashed black line is the least-squares best fit, and the solid black lines indicate ± 1 RMSE.





to approach A1. Further advancements have led to the proliferation of 2D hydraulic models and some 2D bedload transport equations, which aim to account for the proportion of the bed participating in transport and the spatial variation in shear stress (Monsalve et al., 2020). The B2 approach (2D modelled shear stress) that integrates across the frequency distribution of shear stresses did not improve upon approach A1, with an almost identical RMSE (0.36) to approach B1.

The depth-slope method yielded similarly strong correlations between bedload transport and excess shear stress, compared

to the numerically modelled shear stress and 2D approach. This result contradicts a recent proposal by Yager et al. (2018) that highly simplified approaches based on the depth-slope product contribute to significant uncertainties in predicting bedload transport (Gomez and Church, 1989; Barry et al., 2004). More broadly, the ability of the mean shear stress to effectively capture variation in bedload transport is consistent with several lines of evidence. In a re-analysis of data from Oak Creek, OR, Monsalve et al. (2020) compared the Parker and Klingeman (1982) equation to a modified 2D version and found that accounting

for the distribution of shear stresses reduced prediction error by only 13 percent. Their study modelled a range of flows to the same bathymetry, and we obtained a similar result when the bed was allowed to fully adjust to the imposed flow. Using numerical and analytical models, Ferguson (2003) and Francalanci et al. (2012) predicted that variance in shear stress may enhance bedload transport but that this effect rapidly diminishes when $\bar{\tau} \gg \tau_c$. The reason for this sensitivity is the nonlinearity of the bedload transport law, which means that around $\bar{\tau} \approx \tau_c$ small increases in $\tau$ produce relatively large increases in bedload

transport. The similar effectiveness of 1D and 2D functions herein provides empirical evidence that bedload transport is less sensitive to the shape of the shear stress distribution under high relative shear stress conditions

## 4.2 Comparison between 1D and 2D approaches

The four approaches demonstrated key differences based on how shear stress was calculated (depth-slope product vs numerically modelled) and more importantly the formulation (1D vs 2D). Both estimates of mean shear stress were linearly related

to unit discharge but those based on the depth-slope product were 5–23 percent higher (Figure 6), which is consistent with findings by Monsalve et al. (2020). These differences in estimated shear stress led to approximately commensurate differences in the estimated 1D values of $\tau_c^*$ (23 percent higher). Both 1D estimates of $\tau_c^*$ were relatively high for gravel-bed rivers but were within the range of reported estimates from both field and laboratory channels (Buffington and Montgomery, 1997).

Despite having similar prediction errors, the 1D and 2D functions provided considerably different estimates of critical

dimensionless shear stress. Using the 2D approach, estimates of $\tau_c^*$ were 36 and 54 percent higher than the 1D depth-slope and modelled shear stress methods, respectively. In several channels the estimated critical shear stress was greater than the mean shear stress but bedload transport was observed and well predicted by the model (Figure 6), whereas in this case, a 1D equation would predict zero transport. This is a distinct advantage of 2D equations at low flows, as they can account for flows where excess shear stress occupies only a fraction of the bed (Monsalve et al., 2020).

The differences between estimates of $\tau_c^*$ arise from differences in how the equations conceptualise excess shear stress. In a 1D equation, when bedload transport data is available, $\tau_c$ may be back-calculated from the mean shear stress, which is performed here. The value of $\tau_c^*$ is adjusted until excess shear stress explains the observed bedload transport, assuming that $\bar{\tau}$ is responsible for all entrainment. In contrast, the 2D equation does not assume that the mean shear stress participates in



bedload entrainment. Based on the 2D approach, we estimated that the mean shear stress did not exceed the estimated critical
value for the $D_{50}$ until the highest unit discharge experiments, where areas $\tau \leq \bar{\tau}$ occupied only 18–33 percent of the area
experiencing excess shear stress. We did not formally validate these values as we did not anticipate the need for observation,
although the estimates appear reasonable compared to our visual observations of the experiments, and the importance of shear
stresses greater than the mean is also intuitive.

The results suggest that by conceptualising transport as a function of mean shear stress, 1D equations may inflate the
importance of relatively moderate shear stresses and deflate values of $\tau_c^*$. This insight is based on back-calculated values
rather than measurements of incipient motion, although it is important to note that studies measuring incipient motion have
also been based on the mean shear stress and therefore this 1D assumption is subsumed within the results (Gilbert, 1914;
Kramer, 1935; Neill and Yalin, 1969; Wilcock, 1988). The higher estimates of critical dimensionless shear stress using the
2D approach, evaluated by considering the relative importance of shear stresses across the frequency distribution, may have
a stronger conceptual basis. However, it is important to note the fundamental limitations of this approach which still assumes
a double-averaged $\tau_c^*$ and a temporally-averaged distribution of shear stresses, where the mechanistic linkage between shear
stress and bedload movement is tenuous (Yager et al., 2018). Nevertheless, the results highlight that as long as $\tau_c$ is back-
calculated, its value will be highly dependent on how shear stress is estimated, and whether its distribution is treated one- or
two-dimensionally.

Further work is required to investigate differences in 1D and 2D estimates of $\tau_c^*$ under lower excess shear stress conditions.
If broadly applicable, the effectiveness of highly reductionist bedload transport functions based only on median grain size
and mean shear stress would present a convenient assumption for researchers and practitioners interested in channel-forming
flows. More research is required to substantiate this approach under supply-limited conditions and realistic hydrographs that
enable both upward and downward adjustments with inherited channel conditions. Given that our experiments do not allow
for significant lateral adjustment and meandering, the results are most applicable to channels confined by bedrock, or with
cohesive or highly vegetated banks. Fully alluvial channels comprise additional feedbacks that are worthy of investigation, and
the extent to which these affect reach-averaged bedload transport remains poorly understood.

### 4.3 Conclusions

We investigated the performance of 1D and 2D bedload transport functions under high relative shear stress conditions in a
Froude-scaled physical model. The analysis highlights the effectiveness of highly reductionist bedload transport functions,
whereas numerically modelling shear stress to account for flow resistance and energy losses from the channel planform and
banks did not substantially reduce prediction error, nor did accounting for the relative importance of shear stresses across the
frequency distribution. This observation suggests that bedload transport may collapse to a more simple function (i.e., with
mean shear stress and median grain size) under high excess shear stress conditions, provided the data is sufficiently double-
averaged. Given the channels herein have limited lateral mobility, our conclusions are most applicable to channels where lateral
adjustment is suppressed. Further work is required to examine the effect of planform adjustments (widening, meandering),
where small-scale laboratory experiments serve as an effective research tool. The 1D and 2D approaches provided substantially



different estimates of critical dimensionless shear stress, reflecting differences in how these approaches conceptualise excess shear stress. Estimates of $\tau_c^*$ from 2D functions may have a stronger conceptual basis, as they are derived by considering the
relative importance of shear stresses across the frequency distribution, and do not assume that the mean shear stress is sufficient to mobilise the median grain size.

*Code and data availability.*    Raw modelled depth and shear stress values, raw sediment transport data, and summary data from Table 3 are available at Zenodo [DOI: 10.5281/zenodo.6360369] with an Open license (Adams, 2022).

*Author contributions.*    DA was responsible for conceptualisation, data collection, formal analysis, visualisation, and writing. BE was respon-
sible for supervision, review, and editing.

*Competing interests.*    The authors declare that they have no conflicts of interest.

*Acknowledgements.*    We would like to thank Rob Ferguson, Lucy MacKenzie and Will Booker whose suggestions have greatly improved the manuscript. This work was supported by graduate scholarships provided by the Canadian and Australian Governments, and a postgraduate writing-up award (Albert Shimmins Fund) from the University of Melbourne.



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
