# Peer review of "A comparison of 1D and 2D bedload transport functions under high excess shear stress conditions in laterally-constrained gravel-bed rivers: a laboratory study"

_EGUsphere, 2022_

## Referee Comment (RC2)

**Reviewed Manuscript:** A comparison of 1D and 2D bedload transport functions under high excess shear stress conditions in laterally-constrained gravel-bed rivers: a laboratory study

**Authors:** David L. Adams, Brett C. Eaton

**Journal:** Earth Surface Dynamics

**Referee:** Chenge An (anchenge08@163.com)

In this paper, the authors study the bedload transport under relatively high excess shear stress conditions in a laterally-constrained gravel-bed river. More specifically, the authors compare the 1D (based on the reach-averaged shear stress) and 2D (based on the local shear stress) approaches of bedload transport calculation using a Meyer-Peter and Muller type relation. For both approaches, the shear stress are calculated either with the depth-slope product or from a 2D hydraulic model (Nays2DH). Their study finds that under relatively high excess shear stress, both the 1D and 2D approaches (with either way of shear stress calculation) can predict well the bedload transport in the experiments, after some calibration. However, the critical shear stress shows marked difference in the 4 methods (1D vs. 2D, depth-slope product vs numerical modeling), reflecting differences in how these approaches conceptualize excess shear stress.

Results of this study can help deepening our understanding in a traditional topic in the earth surface science: how to calculate the bedload transport in a proper way. The topic is within the scope of *Earth Surface Dynamics*, and the contents are generally well organized. Therefore, I think that this paper could be published on *Esurf* after moderate revision. I list my comments below.

**Main comments**

1. Introduction

Since the topic of this manuscript is about the bedload transport under high excess shear stress conditions (as stated in the title), I think it would be helpful to cover the studies about sheet flow (i.e., bedload layers devolves into a sliding layer of grains that can be several grains thick) in the Introduction. It is likely that the flow condition in this study was not sufficient to induce sheet flow (which often requires a Shields number of 0.5~1.5), but it would still be beneficial to tell the readers what the bedload transport would be like under sufficiently large excess shear stress.

Some literatures about sheet flow are as follows:

Fredsoe, J. and Deigaard, R., 1994, Mechanics of Coastal Sediment Transport, World Scientific, ISBN 9810208405, 369 p.

Gao, P., 2003, Mechanics of bedload transport in the saltation and sheetflow regimes, Ph.D. thesis, Department of Geography, University of Buffalo, State University of New York

Horikawa, K., 1988, Nearshore Dynamics and Coastal Processes, University of Tokyo Press, 522 p.

Parker, G. (2004). 1D sediment transport morphodynamics with applications to rivers and

turbidity currents. (Chapter 7: Relations for 1D bedload transport) http://hydrolab.illinois.edu/people/parkerg//morphodynamics_e-book.htm

Wilson, K. C., 1966, Bed load transport at high shear stresses, *Journal of Hydraulic Engineering*, 92(6), 49-59.

2. Lines 116-118: How is the manning coefficient back-calculated? How do you determine the spatial distribution of manning coefficient? What is the formulation of the Ferguson (2007) relation? As the manning coefficient is one of the most import parameter that determine the flow hydraulics, I would suggest authors to explain in more detail about this content.

3. In the first paragraph of Section 2.2, the authors demonstrated that "Each experimental phase comprises an initial adjustment period during which morphology, hydraulics, and sediment transport are non-stationary. This adjustment period, which may vary from minutes to an hour, is followed by a steady-state period where these characteristics fluctuate around a mean value…In both examples, there is a brief adjustment period with less sediment transport, followed by fluctuations around a mean value."

However, when I looked at Figure 3, I do not clearly observe the two-stage characteristic in the temporal variation of sediment transport rate. I think it would be helpful to do some statistical analysis to justify your demonstration.

4. The experiments applied a widely-graded sediment mixtures, but the MPM type relation based on uniform sediment was applied for the calculation of sediment transport rate. I think that the authors should discuss the effect of multiple grain sizes on the calculation and analysis.

**Specific comments**

1. Line 18: Not only bedload material, but also suspended load, especially for lowland alluvial rivers.

2. Please plot the grain size distribution of the sediment used in the experiment.

3. Lines 87-88: I am not quite sure that I understand this. Maybe it is also not easy for the readers to understand. Please explain more about the measurement frequency.

4. Line 94: "slug" injection. Readers might meet difficulty in understanding the jargon.

5. Caption of Table 3: Are they experimental or model results? I am confused. Also, are they reach-averaged results or results of a certain location.

6. Figure 2: What does the error bar denote? Maximum/minimum or standard deviation?

What does the solid point denote? Mean or median value? Also, does the data in the figure reach-averaged value? Please explain in the caption.

7. Line 140: What do you mean by "second-order processes"?

8. Equation 3: Format problem. Following is what I see in the pdf file. I do not see the integration symbol.

$$q_b \propto \quad^Z (\tau_{(x)} - \tau_{c50})^{1.6} dx/A$$

9. Line 166: Why do you apply a constant slope for the 2D depth-slope method? You can calculate the local slope with the DEM data.

10. Line 174: Is 95 percent a small portion?

11. Lines 187-188: Figure 7a shows the regression of only shear stress, but not the flow depth.

12. Figure 6b: What does the $\tau_{c50}$ in panel b refers to, A1, A2, B1 or B2?

13. Caption of Figure 7b: Do you mean highest (Exp1c(4) and lowest Exp1c(1)? Exp1c(1) has a smaller discharge than Exp1c(4).

14. Conclusions: I suggest to put Conclusions as Section 5, rather than a subsection of Discussion.

---

## Author Response (AR1)

**General response**

We would like to thank the two reviewers for their comments which have greatly improved the manuscript. We respond to each of the reviewers' comments below to address their concerns and outline the improvements to the paper. The improvements have made this work clearer and more accessible, and have helped to better define its contribution to the literature.

**Referee Comment 2**

*R: I think the paper would gain in interest if the authors would provide also a discussion section on the alternative way of using non-threshold approaches (e.g., the Recking, 2013b equation). To my opinion, the whole community studying sediment transport in gravel bed river is still suffering a deep influence of the pioneering works performed in the lab with near-uniform sediment mixtures and narrow flumes (not allowing the geomorphology to develop). In such cases, threshold equations and the concept of threshold of incipient motion make sense to my experience. In natural sites where bedload sediment sizes cover one to two orders of magnitudes and where geomorphology is fully developed, is-it still acceptable to use concepts as threshold for motion and threshold transport equation? I think it does for some targeted tasks, especially for numerical modelling (1D and 2D), thus the interest of this paper. But conceptually non-threshold approaches make more sense to my opinion. This has been quite extensively discussed in the paper of Recking (2013a). Regarding the past publications by the first author, e.g., the paper in Progress in Physical Geography, and his demonstrated capacity to shed a new critical light on the topic of geomorphology, hydraulics and sediment transport, I suggest try adding a section having a broader perspective than to critically assess the way threshold for motion should be approached: what the concept is useful for and what are its limitations.*

A: We agree and have discussed the implications of the findings for non-threshold approaches.

*R: I also suggest trying injecting a bit more of process description in the results section: try going beyond the graph analysis to tell us a story about the channel, the geomorphology and the flows*

A: Agreed. We have added more process description.

*R: Last comment: I find it surprising that although Prof. Eaton wrote very good papers demonstrating that using D50 to assess bedload transport in gravel bed rivers is often irrelevant and that D84 should be preferred, this paper still focus on Tau_c50. Please explain why you did not perform the whole analysis on Tau_c84. Maybe, it is one more argument to shift from threshold equations to go for non-threshold?*

A: We decided to use the D50 as it is more widely used, so this was to increase the reach of the work. By inputting D84, there was little-to-no change in the coefficients or correlations.

*R: L97: I am not sure to understand the ", and every 15 minutes,". Does it means that the mesh buckets were weighted and reintroduced on the conveyor belt every 15 minutes whatever the state of the flume (i.e., drained or not)? Please clarify.*

A: We have clarified "The experiments are best described as pseudo-recirculating as sediment was measured and recirculated at the end of the 5 and 10 minute segments, and for longer segments, every 15 minutes."

*R: L112: why did you use the final DEM and not another one within the three last hours? If this is arbitrary, please just say this.*

A: We have clarified - the selection of the final DEM was arbitrary as any DEM over the steady-state portion of the experiment could have been selected.

*R: L122: I think it is necessary to justify this threshold of 2\*D84 as limit for wetted area. From some basic computations, I understand it corresponds more or less to a Shields stress < 0.05, so it seems acceptable to me but the reader should not have to perform computation to judge whether this threshold is reasonable or not. Please provide your justification.*

A: We have clarified that this is of course defined arbitrarily – the threshold D\*D84 is the saddle between two peaks in the frequency distribution of depths. The mean shear stress is heavily biased if the lower mode not removed. The importance of these shallow regions of flow at the margins is further clarified below.

*R: Figure 2: what exactly are those error bars? The two mm accuracy of the gauge? But these values are mean values over the flume so how much was the standard deviation over the several measurements and how did you propagate uncertainties (epistemic, i.e., 2 mm) and aleatoric (i.e., standard deviation over every gauges)? Please clarify and elaborate on the variability of the measurements at the profiles.*

A: The error bars are computed from measurement precision (2 mm), and so the error bars represent the random error from water depths falling within that range. If we understand correctly, the aleatoric error is not applicable given each gage measures a different part of the channel, i.e., potentially a bar or a pool.

*R: L146: the authors have sufficient data to compute a proxy of the sediment discharge variability to be later used an error bar in the graphs. I suggest computing for instance the standard deviation of the sediment transport of each run on the 3hr period and to add this information in Table 3 as well as in the graphs (e.g., Figure 9).*

A: We have added the standard deviation of output measurements to the table.

*R: L150-170: I suggest to provide full equations for the four ways sediment transport was computed, clearly differencing 2D and 1D and also model and d/S. Consider that for many readers, the uniform flow hypothesis is so deeply assimilated that it might not be clear that Tau is far from being everywhere and always equal to rho \* g \* <S> \* <d>.*

A: We have provided this in an Appendix.

*R: L174: 52%-95% is a "small portion"??*

A: We have corrected this language.

*R: L175: I think it make sense to perform a computation of the unit discharge over only the "wetted area" but a quick control with the 2D modelling is important to my opinion: consider providing the information to the reader that only X-Y% of the water flux pass in the non-wetted area where d<2\*D84. In other contexts, a large share of water is flowing within such shallow flows.*

A: We have provided the information regarding the prevalence of the d < 2\*D84 condition (6.4 mm). Across the flow models, grid cells with flows less than this threshold accounted for 20-63 percent of the channel area where d>0, but only 1-21 percent of the total cross-sectional flow area (mean = 11 percent). This is consistent with visual observations of dispersive/stagnant flow at the channel margins.

*R: L179: A short sentence linking this statistical result with a process interpretation would be appreciated. Am-I correct that it means that at reach-scale, indeed <Tau> ~ rho \* g \* <S> \* <d>?*

A: This is correct, and we have made this statement.

*R: L183-185: I am not sure to fully understand this statement. Could you try helping us to interpret it? Almost no cells comprised condition with Tau_c< tau(x) < <Tau>, and thus almost all wetted cells have Tau(x)>Tau_c? Is it a strong evidence of stress concentration?*

A: We have clarified this section – yes, this is evidence of shear stress concentration.

*R: L198: consider adding something like "i.e., high shear stress and deep flows are close but not at exactly the same locations" to make sure the readers understand. Consider also adding a comment on the effect of morphology on this scattering: would a plane bed morphology present a similar distribution and scattering for instance?*

A: We have clarified this sentence for the reader.

*R: L213: only 20 hours? More than that for the 16+ hrs of experiments at scale 1/25, no? A 20 hr flood is not very long so I suggest using a bigger number to convey a clearer message.*

A: We have clarified: 4-16 hours experimental time or 20-80 hours in the field prototype.

*R: Figure 6: please add a legend of for the black dots to in the right panel.*

A: We have since modified and corrected this figure.

*R: L227: I do not agree with this statement: you compute the mean flow depth or mean shear stress over a river reach using finely tuned 2D model results. An actual 1D approach is, to my experience, rather to measure a transversal profile (or a couple of them) and to compute the flow depth using a uniform flow hypothesis. It is somewhat more similar to the measurement performs at the gauges in the experiments. Consider providing some comments on the differences between mean depth and local or multiple profile measurement or correcting your statement to make it more precise.*

A: Agreed. We have corrected this statement to be more precise.

*Figure 8: contour lines or density shading would be appreciated: in almost all of the interesting region of the plot there is some much overlapping between points that we do not know if a few or thousands of points are plotted.*

A: We have revised this figure to include contours based on 2D kernel density estimates.

*R: L248-251: This topic is also extensively discussed in Recking 2013a.*

A: Reference added.

*R: L263-264: here again, I agree with the statement but the way it is written give me – a non-native speaker, I must confess - the impression that no other approach exist. Pity. What about non-threshold approaches?*

A: We have clarified that the results only directly apply to threshold-based approaches.

**Reviewer 2**

*R: Since the topic of this manuscript is about the bedload transport under high excess shear stress conditions (as stated in the title), I think it would be helpful to cover the studies about sheet flow (i.e., bedload layers devolves into a sliding layer of grains that can be several grains thick) in the Introduction. It is likely that the flow condition in this study was not sufficient to induce sheet flow (which often requires a Shields number of 0.5~1.5), but it would still be beneficial to tell the readers what the bedload transport would be like under sufficiently large excess shear stress.*

A: We appreciate the suggestion but have decided not to include this reference as we do not observe any dynamics approaching sheet flow in our experiments. Perhaps at higher discharge we would observe this process.

*R: Lines 116-118: How is the manning coefficient back-calculated? How do you determine the spatial distribution of manning coefficient? What is the formulation of the Ferguson (2007) relation? As the*

*manning coefficient is one of the most import parameter that determine the flow hydraulics, I would suggest authors to explain in more detail about this content.*

A: We have included the equation that was used to calculate the spatially-variable roughness value, which clarifies the procedure.

*R: In the first paragraph of Section 2.2, the authors demonstrated that "Each experimental phase comprises an initial adjustment period during which morphology, hydraulics, and sediment transport are non-stationary. This adjustment period, which may vary from minutes to an hour, is followed by a steady-state period where these characteristics fluctuate around a mean value…In both examples, there is a brief adjustment period with less sediment transport, followed by fluctuations around a mean value." However, when I looked at Figure 3, I do not clearly observe the two-stage characteristic in the temporal variation of sediment transport rate. I think it would be helpful to do some statistical analysis to justify your demonstration.*

A: We understand the confusion here. In the case of Figure 3a, there is no recorded adjustment period as the channel developed before the first transport measurement was obtained (i.e., 5 minutes in). The transition between adjustment period and steady state is not of interest here and has been addressed in previous publications referenced in this paragraph. We have corrected the expression in the manuscript.

*R: The experiments applied a widely-graded sediment mixtures, but the MPM type relation based on uniform sediment was applied for the calculation of sediment transport rate. I think that the authors should discuss the effect of multiple grain sizes on the calculation and analysis.*

A: Unfortunately, we cannot speculate around the effect of the widely graded sediment as we only collected bulk sediment output. Here, sediment transport was well indexed by the bulk D50.

*R: Line 18: Not only bedload material, but also suspended load, especially for lowland alluvial rivers.*

A: We agree and have corrected this sentence.

*R: Please plot the grain size distribution of the sediment used in the experiment.*

A: We have instead provided a reference to the publication with full information regarding the GSD.

*R: Lines 87-88: I am not quite sure that I understand this. Maybe it is also not easy for the readers to understand. Please explain more about the measurement frequency.*

A: We have slightly modified how this procedure has been expressed to clarify the procedure.

*R: Line 94: "slug" injection. Readers might meet difficulty in understanding the jargon.*

A: We have clarified the meaning of this term.

*R: Caption of Table 3: Are they experimental or model results? I am confused. Also, are they reach-averaged results or results of a certain location.*

A: We have clarified: "Summary of reach-averaged 2D flow model and sediment transport results."

*R: Figure 2: What does the error bar denote? Maximum/minimum or standard deviation?*
*What does the solid point denote? Mean or median value? Also, does the data in the figure reach-averaged value? Please explain in the caption.*

A: We have clarified that these refer to reach-averaged values. The error bars are based on the measurement precision of the stream gages as detailed in the methods.

*R: Line 140: What do you mean by "second-order processes"?*

A: By second order processes we were describing processes that control sediment transport fluctuations rather than the mean transport rate. We have simplified the language here to be clearer.

*R: Equation 3: Format problem. Following is what I see in the pdf file. I do not see the integration symbol.*

$$q_b \propto \int^Z (\tau_{(x)} - \tau_{c50})^{1.6} dx / A$$

A: This may be a pdf problem – we will ensure this is included in the proofs.

*R: Line 166: Why do you apply a constant slope for the 2D depth-slope method? You can calculate the local slope with the DEM data.*

A: We applied a constant slope for the 2D depth-slope product method as this is most analogous to a common field-based cross-section method, in which there is a distribution of flow depths but only a single estimate of local slope.

*R: Line 174: Is 95 percent a small portion?*

A: We have modified the language here.

*R: Lines 187-188: Figure 7a shows the regression of only shear stress, but not the flow depth.*

A: This is correct, although the coefficients for the distribution of modelled flow depths has been included in the manuscript, for brevity we do not present the fitted distribution on this plot. We have better described the figure.

R: Figure 6b: What does the $t_{c50}$ in panel b refers to, A1, A2, B1 or B2?

A: Yes, we have not specified that the tc50 value was for approach B2.

R: Caption of Figure 7b: Do you mean highest (Exp1c(4) and lowest Exp1c(1)? Exp1c(1) has a smaller discharge than Exp1c(4).

A: This was a typo and has been corrected.

R: Conclusions: I suggest to put Conclusions as Section 5, rather than a subsection of Discussion.

A: This was a mistake and has been corrected.

---

## Author Response (AR2)

Response to AE comments.

**General comments**

*E: Referee 2 made a good comment about using D84 rather than D50. Your response to them indicated that using D84 did not change the results, and I think that it might be of interest to readers if you included this in the paper.*

A: In the results, I have clarified this in two ways:

- Noting that changing D50 to D84 has no effect on the correlation between qb and excess shear stress, and
- rather, this change merely reduces the critical dimensionless shear stress values

**Comments by line number (from the tracked changes pdf)**

*E: 9: When reading the abstract before I read the rest of the paper, I didn't follow the sentence starting 'Back-calculated critical…'. Maybe split into two, to start by explaining that you back-calculated tau_c, and then explaining how it varied between the two approaches.*

A: We agree this could be clearer – here is a revised version: "Critical dimensionless shear stress values were back-calculated and were higher for the 2D approach compared to the 1D. This result suggests that 2D critical values account for the relatively greater influence of high shear stresses, whereas the 1D approach assumes that the mean shear stress is sufficient to mobilise the median grain size."

*E: 27: Change to 'widely used'*

A: Done.

*32: Define tau_c50. Explain whether you are referring to spatial or temporal variance in shear stress.*

A: This has been defined, and we have clarified it is the spatial distribution that is being mentioned.

*80: Give the precision of the gauge readings here. The equation for h didn't follow on from the rest of the sentence, as I think that you are measuring h directly, not from the area and w (which was implied by the equation)?*

A: I have provided measurement precision. We are in fact using the h = A/w equation when pairing the gage readings with the topographic data. I believe the order in which we explained this led to confusion and so I have moved the discussion of random error until after the data processing has been explained.

*E: 125: It would be useful to provide a description of how the model calculates shear stress. This would also help later on to explain why in the model the depth and shear stress are not correlated, as in Fig 5.*

A: We agree and have added that in the numerical model local shear stress is calculated using the bed friction coefficient and depth-averaged flow velocity components.

*E: Table 3: Clarify which variables are modelled and which are measured in the flume.*

A: This has been clarified in the table caption: "Summary of reach-averaged hydraulics (from 2D flow model) and sediment transport (from measurements)."

*E: 172: The different methods are a key point in this paper, and could still be more clearly explained. It's up to you, but I'd move the appendix material to here, as it isn't long, but is important. Two things weren't clear to me. First, are all the depths in the depth-slope approach also derived from the flow modelling? Or the gauge measurements? I assume that it's the former, but when you specify that your second shear stress measurements come from the flow modelling, it implies that the first approach might come from a different data set. Secondly, it wasn't clear to me at what spatial resolution you were calculating depth and shear stress in the 2D approach – at the 1 mm resolution of the DEM?*

A: We have decided to keep the full equations in the appendix as the existing explanation in the methods is self-contained.

We agree this was ambiguous and have been more clear – "1) the depth-slope product ($\tau = \rho gdS$) based on numerically modelled flow depths, and 2) numerically modelled shear stresses"

In the methods we specified that the numerical models had a grid resolution of 5 mm. For clarity, I've added a second mention of this resolution when discussing the frequency distributions of shear stress in the results section.

*E:* Table 4: I think that this should be in results, rather than the methods.

A: It is now in the results section.

*E:* 197: This comment about the amount of the bed where tau is between tau_c50 and mean tau comes up a few times, but I didn't quite understand why it was so important. A reference to Fig 6 here would help. 37% of the area doesn't sound that insignificant. Would it also help to explain what proportion of the bed had tau greater than mean tau?

A: We have added the reference to Figure 5. We have altered some of the interpretation in the discussion to make our point clearer: that although the mean shear stress may particulate in bedload transport, it may be far less important than the higher shear stresses. This is intuitive, but it is important to note as an explanation for why the back-calculated critical dimensionless shear stress values may always be higher if they come from 2D transport functions.

*E:* 208: I assume that you mean modelled shear stress?

A: We have clarified this.

*E:* 210: I don't think that you have defined how you normalised depth and shear stress.

A: We have made sure to use the term "mean-normalised" rather than just "normalised".

*E:* Fig 6: Add the symbols q and tau to first sentence of the caption.

A: Done.

*E:* Fig 7: x axis label in panel a should be tau/mean tau or depth/mean depth.

A: I've made the correction.